



**1  Modelling the effect of submarine iceberg melting on glacier-adjacent**

**2  water properties**

**3  Benjamin Davison[1,2], Tom Cowton[1], Andrew Sole[3], Finlo Cottier[4,5], Pete Nienow[6]**

[1]Department of Geography and Sustainable Development, University of St Andrews, St Andrews, UK
[2]School of Earth and Environment, University of Leeds, Leeds, UK
[3]Department of Geography, University of Sheffield, Sheffield, UK
[4]Scottish Association for Marine Science, Scottish Marine Institute, Oban, UK
[5]Department of Arctic and Marine Biology, UiT The Arctic University of Norway, Tromsø, Norway
[6]School of Geosciences, University of Edinburgh, Edinburgh, UK
Correspondence email: b.davison@leeds.ac.uk
**Abstract**
The rate of ocean-driven retreat of Greenland's tidewater glaciers remains highly uncertain in
predictions of future sea level rise, in part due to poorly constrained glacier-adjacent water properties.
Icebergs and their meltwater contributions are likely important modifiers of fjord water properties, yet
their effect is poorly understood. Here, we use a 3-D ocean circulation model, coupled to a submarine
iceberg melt module, to investigate the effect of submarine iceberg melting on glacier-adjacent water
properties in a range of idealised settings. Submarine iceberg melting can modify glacier-adjacent water
properties in three principle ways: (1) substantial cooling and modest freshening in the upper ~50 m of
the water column; (2) warming of Polar Water at intermediate depths due to iceberg melt-induced
upwelling of warm Atlantic Water, and; (3) warming of the deeper Atlantic Water layer when vertical
temperature gradients through this layer are steep (due to vertical mixing of warm water at depth), but
cooling of the Atlantic Water layer when vertical temperature gradients are shallow. The overall effect
of iceberg melt is to make glacier-adjacent water properties more uniform with depth. When icebergs
extend to, or below, the depth of a sill at the fjord mouth, they can cause cooling throughout the entire
water column. All of these effects are more pronounced in fjords with higher iceberg concentrations
and deeper iceberg keel depths. These iceberg melt-induced changes to glacier-adjacent water
properties will reduce rates of glacier submarine melting near the surface, but increase them in the Polar
Water layer, and cause typically modest impacts in the Atlantic Water layer. These results characterise
the important role of submarine iceberg melting in modifying ice sheet-ocean interaction, and highlight
the need to improve representations of fjord processes in ice sheet-scale models.



## 1. Introduction

Predicting the rates of ocean-driven retreat of Greenland's tidewater glaciers remains one of the largest uncertainties in estimating future sea level rise (Edwards et al., 2021; Meredith et al., 2020). This uncertainty is partly due to limited constraints on the ocean-driven thermal forcing of tidewater glacier calving fronts, which reflects in part the difficulty in obtaining hydrographic observations in the proximity of tidewater glacier termini (Jackson et al., 2017, 2020; Sutherland et al., 2019). The few observations of water properties in the inner part of glacial fjords demonstrate that there are typically substantial differences between glacier-adjacent water properties and those near the fjord mouth (e.g. Inall et al., 2014; Jakobsson et al., 2020; Straneo et al., 2011), indicating that substantial modification of water temperature and salinity can occur within glacial fjords. Due to the relatively small number of observations and insufficient model constraints on glacier-adjacent water properties, ice sheet models used to simulate glacier retreat must be forced with far-field (i.e. acquired on and beyond the continental shelf) ocean boundary conditions that do not include fjord-scale influences (Goelzer et al., 2020; Slater et al., 2019), thereby introducing uncertainty into the resulting projections of ice sheet mass loss.

Glacier-adjacent water properties can differ from those near the fjord mouth for several reasons. Meltwater runoff enters the fjord at depth where tidewater glaciers meet the ocean ('subglacial discharge'). In Greenland's fjords, warm water of Atlantic origin (Atlantic Water, AW) is generally found at depth, whilst colder, fresher water of Polar origin (Polar Water, PW) is found at intermediate depths (Straneo and Heimbach, 2013; Sutherland and Pickart, 2008). The cold, fresh subglacial discharge is buoyant when it enters the fjord, so rises as a turbulent plume (Jenkins, 2011). As it rises, it entrains fjord water, which mixes with the subglacial discharge as it ascends towards the fjord surface (e.g. Beaird et al., 2018). In this way, subglacial discharge-driven plumes act as mixing engines at the head of glacial fjords. Due to the temperature stratification in Greenland's fjords, plumes at deeply-grounded glaciers (i.e. deeper than the PW-AW interface) often draw the relatively warm AW towards the fjord surface, thereby warming surface and near-surface waters (e.g. Carroll et al., 2016; Straneo et al., 2010, 2011). In contrast, plumes at shallowly-grounded glaciers can cause cooling at and near the fjord surface, as cold subglacial discharge and entrained PW is upwelled into surface layers that are seasonally warmed by solar radiation (Carroll et al., 2016). Models that include glacial plumes are able to reproduce these effects convincingly (Carroll et al., 2016; Cowton et al., 2015; Jackson et al., 2017). However, there remain substantial differences between modelled water properties and those that are observed adjacent to tidewater glaciers (Cowton et al., 2016; Davison et al., 2020; Fraser and Inall, 2018).

Several recent studies have identified icebergs as a substantial freshwater source in some of Greenland's fjords, with iceberg freshwater volumes comparable to or greater than ice sheet runoff (Enderlin et al., 2016, 2018; Jackson and Straneo, 2016; Moon et al., 2017; Moyer et al., 2019; Rezvanbehbahani et al., 2020). Furthermore, modelling of one of these fjords suggests that including the heat and salt fluxes



associated with submarine iceberg melting increases greatly the model's ability to reproduce observed
glacier-adjacent water properties (Davison et al., 2020). However, iceberg concentration, keel depth,
and size-frequency distribution likely vary hugely between fjords as well as over time, though
observations of icebergs at the fjord scale are sparse (Enderlin et al., 2016; Moyer et al., 2019;
Rezvanbehbahani et al., 2020; Sulak et al., 2017). As such, it is likely that the effect of icebergs on
glacier-adjacent water properties will also vary both spatially (i.e. between fjords) and temporally. This
variability likely results in different thermal forcing of tidewater glaciers for a given set of far-field
ocean conditions. Constraining the effect of icebergs on glacier-adjacent water properties, and thus
glacier submarine melt rates, is therefore a necessary step in order to improve projections of ice sheet
mass loss.
Here, we use an ocean circulation model in a series of idealised fjord-scale simulations to examine how
icebergs affect glacier-adjacent water properties across a range of Greenland-relevant scenarios. We
first consider how iceberg concentration, keel depth and size-frequency distribution individually affect
glacier-adjacent water properties. We then consider a range of representative iceberg and ocean
scenarios, to examine how these parameters interact to determine water properties in the critical region
adjacent to tidewater glacier termini. Greenland's fjords are complex and varied in their geometry,
ranging from short, narrow inlets to those that are long and wide, each with varying sinuosity and
bathymetry, and often with several tributaries and sills of varying depth along their length. It would be
impractical to attempt to characterise all of these systems. Therefore, we focus here on two simple fjord
geometries: one with no sills and another with a single entrance sill, which we expect to be of particular
importance for iceberg-ocean interaction given the capacity of sills to concentrate fjord-shelf water
exchange near the surface where icebergs are concentrated.

## 2. Methods

### 2.1. Model domain

We use the Massachusetts Institute of Technology general circulation model (MITgcm) in its non-
hydrostatic configuration (Marshall et al., 1997a, 1997b) to model submarine ice melting and circulation
in an idealised fjord 50 km in length and 5 km in width. In most simulations, the domain is uniformly
500 m deep. However, in some simulations, we include a sill which limits the overlying water depth to
100 m (uniform across the entire width of the fjord, and approximately 5 km wide in the along-fjord
direction, with a Gaussian profile), centred 10 km from the open boundary (Fig. 1a). Model resolution
is uniformly 500 m horizontally and 10 m vertically. The fjord sides are closed boundaries, while at the
open ocean boundary we impose a 5 km sponge layer, in which conditions are relaxed towards those
imposed at the boundary (e.g. Cowton et al., 2016; Sciascia et al., 2013; Slater et al., 2015). The glacier-
end of the domain is closed and consists of a virtual ice wall 5 km wide and 500 m high. In simulations



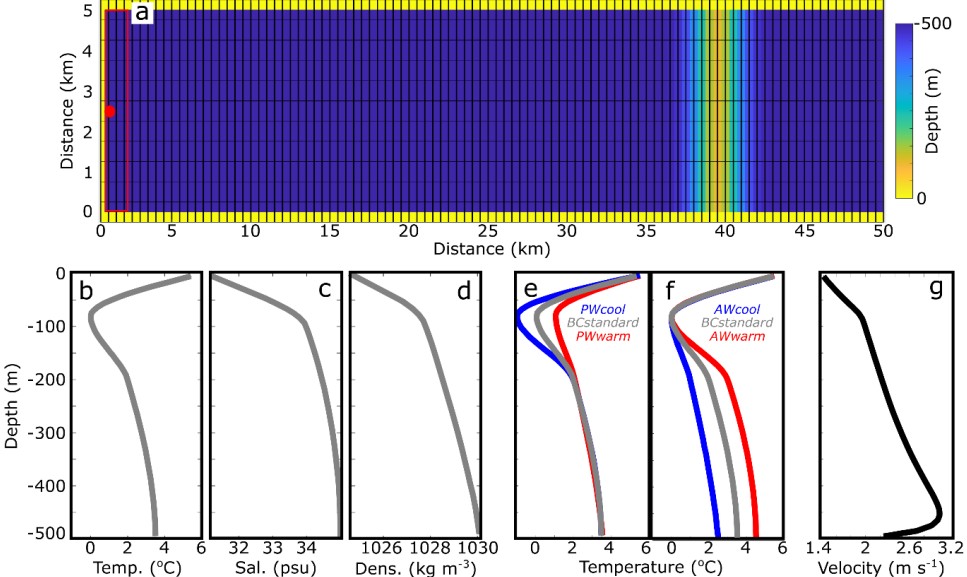

Figure 1. Model domain and boundary conditions. (a) Plan-view of model bathymetry with sill, with the ice wall at the left end of the domain (0 km) and the open boundary on the right. Hatching indicates model resolution (note that grid cells are 500 m x 500 m in the horizontal). The red dot marks the location of subglacial discharge injection and the red box indicates the region from which steady-state glacier-adjacent water properties were extracted. In simulations without a sill, the domain is uniformly 500 m deep. Vertical profiles of (b) temperature, (c) salinity and (d) density with *BCstandard*. (e) Temperature profiles with varying PW temperature. (f) Temperature profiles with varying AW temperature. (g) Example plume vertical velocity from the simulation with iceberg scenario five, 500 m³ s⁻¹ runoff and *BCstandard* boundary conditions.

incorporating runoff, this is input at a rate of 500 m³ s⁻¹, a value typical of many of Greenland's tidewater
glaciers (Mankoff et al., 2020), at the centre of the base of the ice wall (Fig. 1a). The velocity of the
runoff-driven plume (e.g. Fig. 1g) and the melting of the ice wall were calculated using the 'IcePlume'
package (Cowton et al., 2015). In common with several previous studies (Kimura et al., 2014; Slater et
al., 2015; Xu et al., 2013), we implement a free slip condition on the fjord walls and ice front and do
not simulate the effects of sea ice, atmospheric forcing or tides.
**2.2. Initial and open boundary conditions**
We use idealised representations of temperature and salinity profiles commonly observed at the mouth
of Greenland's south-eastern fjords during late-summer as initial and open boundary conditions
(Sutherland et al., 2014). In our standard setup, this idealised profile is a cubic interpolation between
6°C and 31 psu at the fjord surface, 0°C and 34 psu at 100 m depth, 2°C at 200 m and 3.5°C at 500 m
depth, where salinity is greatest at 35 psu (Fig. 1b-d). In this way, the upper several tens of metres
represent waters that are seasonally warmed by solar insolation, whilst the relatively cold intermediate



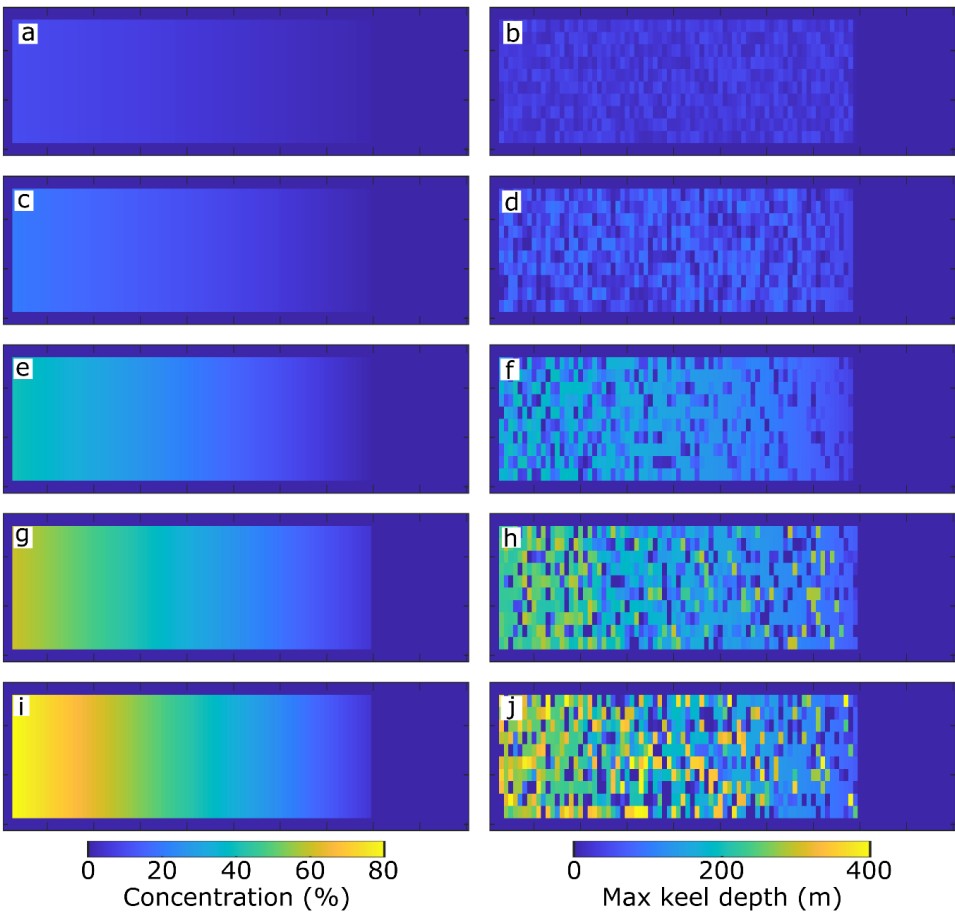

Figure 2. Iceberg concentration (left column) and maximum iceberg keel depth (right column) for iceberg scenarios one to five (top to bottom). All panels show the domain in plan-view, and are 50 km long and 5 km across.

layer, centred 100 m below the fjord surface, represents the PW layer, which is underlain by warmer,
more saline water representing the AW layer. Henceforth, we refer to this set of boundary conditions
as *BCstandard*. In separate simulations, we use temperature minima at 100 m of -1°C (*PWcool*) and
1°C (*PWwarm*) and temperature maxima at 500 m of 2.5°C (*AWcool*) and 4.5°C (*AWwarm*) (Fig. 1e,f).
Changing the temperature of the AW and PW layers causes corresponding changes in the vertical
temperature gradient (Fig. 1e,f), the effects of which are discussed in Sect. 3.2. Initial and open
boundary salinity is kept constant between simulations, but density changes between simulations are
negligible. Boundary conditions were kept constant throughout each simulation.



Table 1. Details of each iceberg scenario. Concentration is the percentage of the fjord in plan-view occupied by icebergs. Iceberg concentration was linearly interpolated from the maximum value (adjacent to the glacier wall) to the minimum value 40 km down fjord.

| Iceberg scenario | Max. draught (m) | Exponent | Concentration [max,min] (%) | Surface area (km²) |
|---|---|---|---|---|
| Scenario 1 | 50 | 1.6 | [10,1] | 44.5 |
| Scenario 2 | 100 | 1.7 | [20,1] | 76.5 |
| Scenario 3 | 200 | 1.8 | [40,1] | 141 |
| Scenario 4 | 300 | 1.9 | [60,5] | 235 |
| Scenario 5 | 400 | 2.1 | [80,5] | 316 |

### 2.3. Iceberg-ocean interaction

Submarine iceberg melting is simulated using the 'IceBerg' package within MITgcm (Davison et al., 2020), with an ice temperature of -10°C (Inall et al., 2014; Luthi et al., 2002; Sciascia et al., 2013; Sutherland and Straneo, 2012). This package uses the velocity-dependent three-equation melt rate parameterisation (Holland and Jenkins, 1999; Xu et al., 2012) to resolve the vertical pattern of submarine melting of individual icebergs. The temperature and salinity fluxes associated with melting of individual iceberg faces within a grid cell are calculated based on local temperature, salinity and face-normal velocity. Face-normal current speed is calculated assuming that icebergs drift with the average current velocity along their draught (though we note that the iceberg locations are kept constant through each simulation). Melt-driven plumes are not simulated directly; instead, their effect on melt rates is parameterised by applying a minimum face-normal current speed of 0.06 m s⁻¹ to each iceberg face. This minimum current speed is based on line plume modelling (Davison et al., 2020). The package does not include the effect of waves or mechanical iceberg breakup; therefore, melt rates calculated here are conservative. We use standard parameter values (Cowton et al., 2015; Davison et al., 2020; Jackson et al., 2020) for the drag coefficient (0.0025), and thermal and salt turbulent transfer coefficients (0.022 and 0.00062, respectively). The icebergs are rectangular in plan-view and have flat, vertical sides. All icebergs have length, $l$, to width ratios of 1.62:1 (Dowdeswell et al., 1992), and iceberg keel depth, $d$, is related to iceberg length through, $d=2.91l^{071}$ (Barker et al., 2004).

In Sect. 3.1, we consider a range of iceberg concentrations, maximum keel depths and size-frequency distributions, whilst using only the *BCstandard* boundary conditions. In all setups, iceberg concentration is uniform across the fjord and decreases linearly from a maximum adjacent to the virtual ice wall to a minimum 10 km from the open boundary. In Sect. 3.1, iceberg concentration (defined as the percentage of the fjord surface in plan-view occupied by icebergs), is 80% adjacent to the ice wall and decreases to 5% in our *c1* experiment, and is reduced to 75, 50 and 25% of these values in our *c0.75*, *c0.5*, and *c0.25* experiments, respectively. Regardless of concentration, we use a maximum iceberg keel depth of 300 m and the size-frequency distribution of the icebergs is described using a



power law with an exponent of -2, which is similar to that observed in Sermilik Fjord (Sulak et al.,
2017). In separate simulations, we assign maximum iceberg keel depths of 50 m, 150 m, 250 m, 350 m
and 450 m, whilst maintaining the $c1$ concentration and the -2 power law exponent. We then vary the
size-frequency distribution power law exponent from -1.6 to -2.1 in increments of 0.1 (covering the
range observed to date in Greenland's fjords (Rezvanbehbahani et al., 2020; Sulak et al., 2017)), whilst
retaining the $c1$ concentration and the 300 m maximum keel depth.
In Sect. 3.2 onwards, we consider five realistic combinations of iceberg concentration, maximum
iceberg keel depth and power law exponent, in order to approximate the range of iceberg geometries
and distributions found in Greenland's fjords (Fig. 2). These iceberg setups range from those
representing a fjord hosting few and small icebergs, such as Kangerlussuup Sermia Fjord (Sulak et al.,
2017) (scenario one), to those representing an iceberg-congested fjord, such as Sermilik Fjord (scenario
five) (Fig. 2; Table 1).

### 3. Results

#### 3.1. The effect of iceberg concentration, keel depth and size-frequency distribution on glacier-adjacent water properties

The effect of iceberg melt on glacier-adjacent water properties depends on iceberg geometry, iceberg
concentration and iceberg size-frequency distribution (Fig. 3), as well as on the presence or absence of
subglacial discharge. In the absence of subglacial discharge, icebergs modify glacier-adjacent water
properties (here defined as the average properties of the water within 2 km of the ice wall; Fig. 1a) in
two main ways. Firstly, they cause substantial (6-7.5°C) cooling in the upper ~60 m of the water column,
relative to the initial conditions (Fig. 3a-c). The amount of cooling in this near-surface layer depends
somewhat on iceberg concentration, with steady-state water temperature varying between ~-1.5°C and
~0°C over the range of iceberg concentrations considered, but is otherwise relatively insensitive to
changing iceberg geometry and distribution (Fig. 3a-c). Secondly, warming of up to ~1°C occurs below
~80 m because iceberg melting causes localised freshening at depth. The resulting iceberg melt-
modified water (i.e. the mixture of iceberg freshwater and ambient water at depth) is less dense than the
surrounding water and rises buoyantly towards the fjord surface. The vertical extent and magnitude of
the resulting warming generally increase with maximum iceberg keel depth (Fig. 3b), because icebergs
with deeper keels cause upwelling of deeper AW (which in this case is also warmer (Fig. 1b)). This
warming effect does not extend to the fjord surface, because the stronger stratification near the surface
limits upwelling and because iceberg-ocean contact areas are much greater near the surface, so cooling
due to localised iceberg melting dominates. When subglacial discharge is included, the effect of iceberg
melt on glacier-adjacent water properties at depth (below 60 m) is similar to that in simulations without
subglacial discharge, but glacier-adjacent water temperatures in the upper ~60 m of the water column
display a greater range and the cooling of the near-surface waters is considerably reduced (Fig. 3d-f).

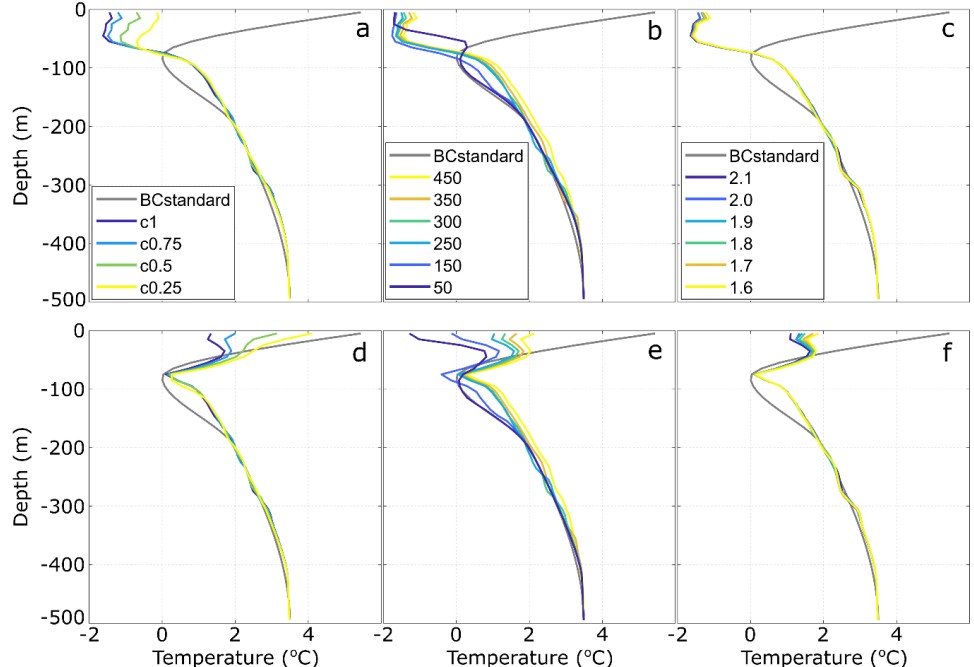

Figure 3. Glacier-adjacent water temperature vs iceberg geometry and distribution. Effect of iceberg concentration (a & d), maximum iceberg draught (b & e) and exponent describing the size-frequency distribution (c & f). Panels (a-c) are for simulations without runoff, whilst panels (d-f) are for simulations with 500 $m^3$ $s^{-1}$ runoff.

This is because the runoff causes strong upwelling of AW towards the fjord surface and increases rates
of fjord-shelf exchange, which counters some of the iceberg-induced cooling of near-surface waters.

**3.2. Combining iceberg scenarios and ocean conditions**
In reality, changes in iceberg concentration, keel depth and size-frequency distribution do not occur in
isolation and there are characteristic relationships between those iceberg descriptors (Sulak et al., 2017).
Fjords hosting large glaciers, such as Sermilik Fjord and Helheim Glacier in east Greenland, tend to
contain both high iceberg concentrations and large, deeply-draughted icebergs, whilst those with lower
iceberg concentrations, such as Kangerlussuup Sermia Fjord, also tend to contain smaller icebergs. To
better represent the range of iceberg conditions found in Greenland's fjords, we consider five iceberg
'scenarios' (Fig. 2; Table 1), ranging from a fjord with low iceberg concentration, shallow iceberg keels
and fairly uniform iceberg sizes (iceberg scenario one), to a fjord with high iceberg concentration, deep
iceberg keels and a large range of iceberg sizes (iceberg scenario five). For each of these scenarios, we
examine steady-state glacier-adjacent water temperature for a range of ocean boundary conditions, and



with and without a shallow (100 m) sill. We therefore consider three different PW and AW temperatures
in turn (Fig. 1e,f), and examine the resulting glacier-adjacent water properties for each of the five
iceberg scenarios. To isolate the effect of iceberg melting from other processes, we compare each of the
above simulations to identical simulations without icebergs.

### 3.2.1.   Changing Polar Water temperature

Fig. 4 shows steady-state glacier-adjacent water properties for the range of iceberg scenarios and PW
temperatures considered. In all iceberg scenarios, there is substantial (~2°C or more) cooling in the
upper ~60 m, with greater cooling in scenarios with higher iceberg concentrations. Other than this near-
surface cooling, glacier-adjacent water properties are very similar to open ocean conditions in iceberg
scenarios one and two (which have the lowest iceberg concentrations; Fig. 2; Table 1). However, in
iceberg scenarios three to five, the PW layer is increasingly modified (Fig.s 4c-e). With *PWcool*,
icebergs in these scenarios cause on average a net *warming* of 1.02°C in the 80-200 m depth range,
compared to simulations without icebergs. Conversely, with *PWwarm*, the icebergs cause a net cooling
of 0.30°C over the same depth range, such that the steady-state temperature profiles for both sets of
initial conditions (*PWcool* and *PWwarm*) are similar. With *BCstandard*, the influence of icebergs on
glacier-adjacent water properties falls between the two, with the net effect being a slight (0.43°C)
warming (Fig. 4c-e). These changes arise due to differing balances between cooling due to iceberg
melting, and warming due to buoyancy-induced upwelling of relatively warm AW water. With *PWcool*
there is relatively little iceberg melting in the PW layer (because the PW is close to the *in-situ* freezing
point), and so warming due to upwelling of AW dominates (driven by iceberg melting at greater depth
in the warmer AW layer). In contrast, with *PWwarm*, iceberg melt rates in the PW layer are
comparatively high, and the temperature difference between the PW and AW layers is reduced, so
localised cooling offsets warming due to turbulent upwelling. In short, under the conditions represented
by these simulations, submarine iceberg melting acts to make glacier-adjacent water temperature more
uniform with depth (Fig. 4c-e).
The addition of a 100 m deep sill near the fjord mouth serves to amplify the cooling effect of icebergs
(Fig. 4f-j). Sills typically block external shelf waters below the sill depth from entering the fjord (unless
external forcing causes a shallowing of isopycnals seaward of the sill), causing the fjord basin bounded
by the sill to be replenished by waters sourced only from above the sill depth (e.g. Jakobsson et al.,
2020). When icebergs reach down to the sill depth, all water entering the fjord may thus be subject to
melt-driven cooling. The result is that icebergs cause cooling throughout the water column, even below
the deepest iceberg keels and below the sill depth (Fig. 4f-j). This cooling is increasingly pronounced
as the PW temperature increases and with more concentrated and deeper icebergs (Fig. 4f-j).  For
example, over the 100 to 500 m depth range with *PWcool*, icebergs cause 0.21°C cooling on average in

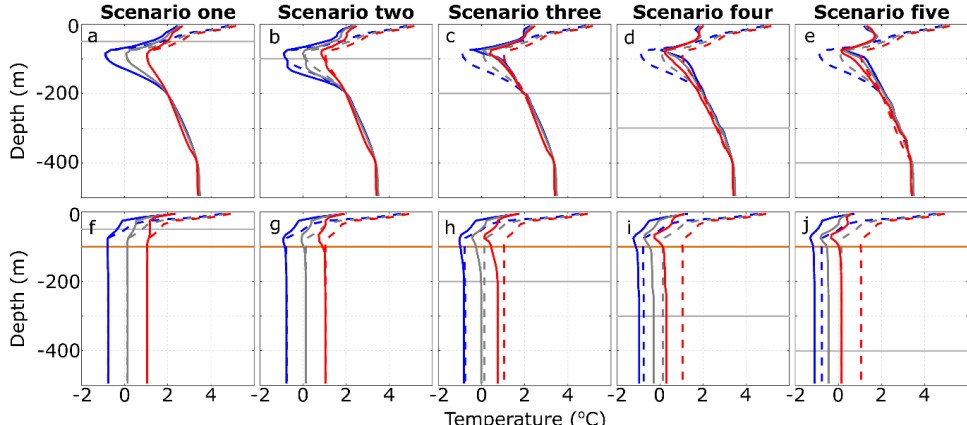

Figure 4. Steady-state glacier-adjacent water temperature for a range of initial Polar Water conditions. In all plots, solid and dashed lines indicate simulations with and without icebergs, respectively. Plots a-e show configurations with a flat-bottomed domain, whilst f-j show those with a 100 m deep sill. Grey, blue and red lines show scenarios using the *BCstandard*, *PWcool* and *PWwarm* boundary conditions respectively (shown in Figure 1e). The horizontal grey lines indicate the maximum iceberg keel depth in each scenario, and the horizontal orange lines in panels f-j indicate the sill depth.

iceberg scenarios three to five (0.06°C in scenario three and 0.35°C in scenario five); whilst with
*PWwarm*, icebergs cause 0.67°C cooling on average (0.33°C in scenario three and 0.91°C in scenario
five).
The varied effects of icebergs on glacier-adjacent water properties are apparent in temperature-salinity
space (Fig. 5). Initial glacier-adjacent water properties are inherited from those prescribed at the fjord
mouth; however, icebergs modify fjord waters through ice melt and meltwater-driven vertical mixing.
Comparing temperature-salinity profiles of simulations with and without icebergs illustrates these
effects (Fig. 5). In the upper ~60 m of all simulations with icebergs, iceberg melting causes substantial
cooling and slight freshening (e.g. compare solid and open circles in Fig. 5 – solid circles are drawn
down and slightly left in temperature-salinity space). Deeper in the water column (below 100 m), the
influence of iceberg melting on water properties depends on the iceberg scenario and the presence or
absence of a sill. In iceberg scenario one (Fig. 5a, b), iceberg melting causes very little modification of
waters below 100 m, even in the presence of a sill (Fig. 5b). This is because the icebergs do not extend
to the sill water depth and so there is some unmodified exchange between the fjord and shelf. In iceberg
scenario five, icebergs cause on average 0.19°C warming of waters below 100 m when there is no sill,
and cooling of 0.61°C below 100 m when there is a sill (Fig. 5b). This cooling below the maximum
iceberg draught occurs in all iceberg scenarios in which icebergs extend to sill depth, but is most
apparent in the higher iceberg concentration scenarios (e.g. Fig. 5d). The simulated changes in water

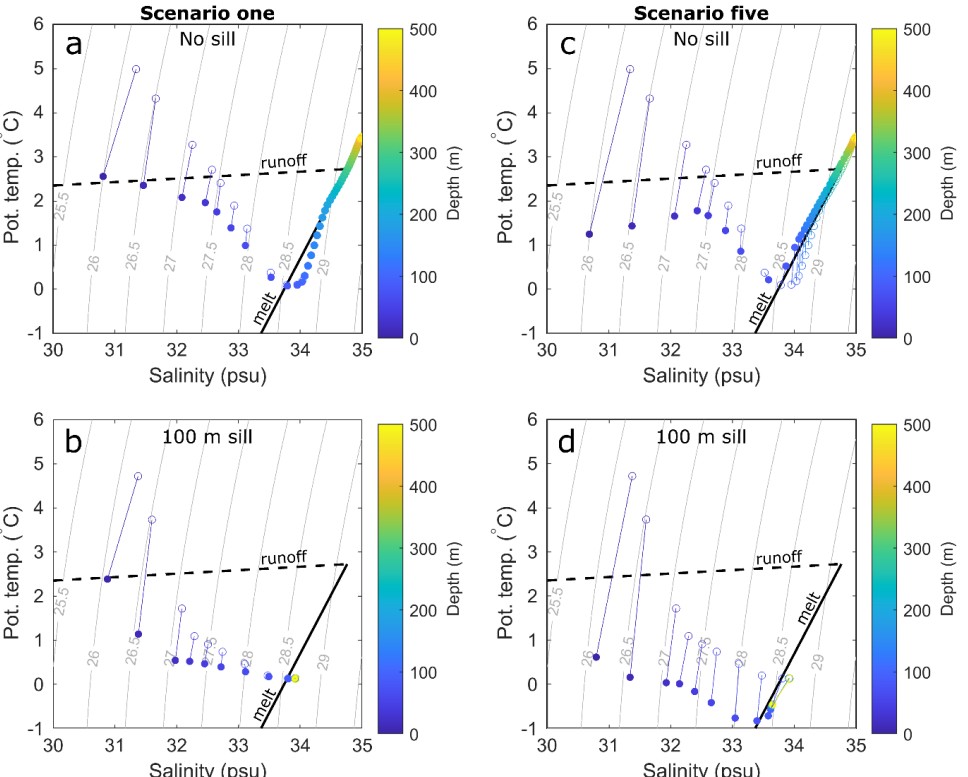

Figure 5. Glacier-adjacent temperature and salinity with (solid circles) and without icebergs (open circles) for various iceberg and sill scenarios and with *BCstandard* boundary conditions. Panels (a) and (b) show iceberg scenario one without a 100 m sill (a) and with a sill (b). Panels (c) and (d) show iceberg scenario five, without a sill (c) and with a 100 m sill (d). Solid lines joining open and closed circles indicate connected data points extracted from the same model depth.

properties arise due the combined effects of local iceberg melting and fjord circulation. Submarine
iceberg melting reduces the density of surrounding waters, causing upwelling until those waters
equilibrate at a new neutral buoyancy depth with respect to the fjord stratification. Within the
temperature-salinity space of Greenland's fjords, density is predominantly salinity controlled.
Therefore, the salinity stratification is little changed by iceberg melting, whilst the temperature changes
are much more pronounced. This means that the iceberg melt-induced migrations through temperature-
salinity space that are often steeper than predicted by the submarine melt mixing line (Gade, 1979).

**3.2.2. Changing Atlantic Water temperature**
We also examine the interactions between iceberg scenarios and changes to AW temperature (Fig. 6).
As in the PW scenarios, there is always marked cooling in the upper ~60 m of the water column and


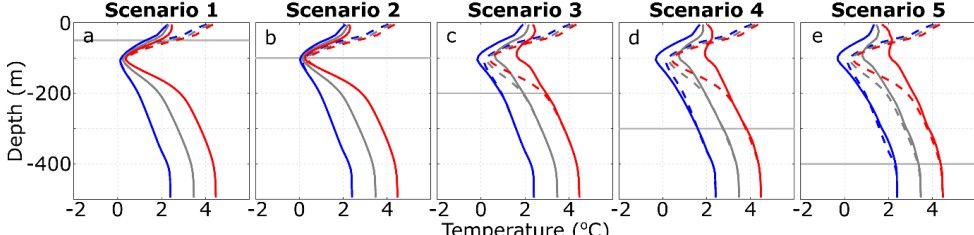

Figure 6. Steady-state glacier-adjacent water temperature for a range of initial Atlantic Water conditions and with a flat-bottomed domain. In all plots, solid and dashed lines indicate simulations with and without icebergs, respectively. Grey, blue and red lines show scenarios using the *BCstandard*, *AWcool* and *AWwarm* boundary conditions, respectively (shown in Figure 1f). The horizontal grey lines indicate the maximum iceberg keel depth in each scenario.

water modification below this is minimal for iceberg scenarios one and two. In iceberg scenarios three
to five, icebergs penetrate to a greater depth and thus into the AW layer, releasing freshwater which
causes upwelling of AW. In these cases, the net effect of icebergs on water properties between ~80 m
and the maximum iceberg keel depth depends on the balance between cooling due to localised iceberg
melting, and warming due to upwelling of AW. With *AWwarm*, there is a steep temperature gradient
between the cold PW and warmer AW layers. Consequently, upwelling of AW causes notable warming
in the PW layer that offsets localised iceberg-induced cooling. In the scenarios with greater iceberg
concentration (e.g. iceberg scenario five; Fig. 6e), the icebergs penetrate deeper into the AW layer and
so can induce upwelling of the deeper, warmer water, resulting in more warming and over a greater
depth range than in the lower iceberg concentration scenarios. However, with *AWcool*, the vertical
temperature gradient is reduced, so cooling due to localised iceberg melting dominates the signal
between the maximum iceberg draught and ~80 m.
This dependence of iceberg modification of glacier-adjacent water properties on the temperature
gradient through the AW layer is further illustrated by sensitivity tests in which the temperature of the
AW layer was modified in two ways relative to *BCstandard*. First, to examine whether the absolute
temperature of the water column affected the balance between upwelling and melting, the entire water
column was uniformly warmed by 1°C. With this uniform shift in temperature, the pattern of
temperature with depth is similar to that of *BCstandard* (compare dashed grey and red lines in Fig. 7b),
illustrating that the additional upwelling-driven warming with *AWwarm* is due to the steeper
temperature gradient between the PW and AW layers, rather than the absolute temperature of the AW.
Secondly, to illustrate the importance of the temperature gradient within the AW layer, we made the
AW layer uniformly 3.5°C. With this set of boundary conditions, upwelling-driven warming dominates
in the PW layer, because of upwelling of warm AW, whilst melt-driven cooling dominates in the AW
layer because upwelling-driven warming is muted (Fig. 7c). Thus, the average warming below ~80 m


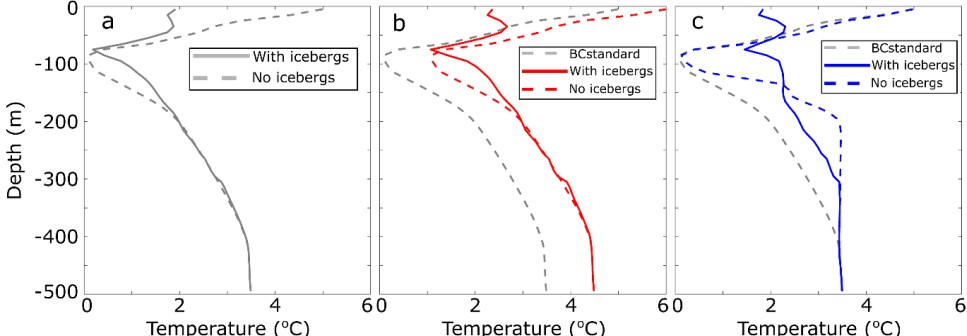

Figure 7. AW temperature gradient sensitivity tests. Panels show simulations using (a) *BCstandard*, (b) temperature profile shifted by 1°C throughout the water column, and (c) uniform initial AW temperature of 3.5°C. Steady-state conditions without icebergs using *BCstandard* (grey line) are also shown in (b) and (c) for reference.

that we simulate with *AWwarm* is strongly sensitive to the vertical temperature gradient, and not only
the average or maximum temperature of the AW.
With the addition of a 100 m sill, AW does not propagate into the fjord under the conditions simulated
here. Thus in steady-state, glacier-adjacent water properties are unaffected by AW and adopt the
properties of the PW layer (modified by iceberg melting and runoff). The resulting profiles therefore
resemble the dashed pale blue lines in Fig. 4f-j and are not shown here.

**4. Discussion**
**4.1. Comparison with observations and applicability to real fjords**
Our simulations suggest that several changes to glacier-adjacent water properties can occur due to
submarine iceberg melting. In almost all simulations, we simulate pronounced (>2°C) cooling in the
upper several tens of metres of the water column. Deeper in the water column (between ~80 m and the
maximum iceberg keel depth), both iceberg-induced cooling and warming can occur (e.g. Fig. 4 and 6),
depending on the balance between cooling due local iceberg melting and warming due to melt-driven
upwelling. The balance between these processes depends on the iceberg contact area at depth available
for local melting (and therefore cooling) and on the temperature of the upwelling water. When vertical
temperature gradients are steep (e.g. with *AWwarm*; Fig. 6), icebergs can cause warming between their
maximum keel depth and the surface layer. This is particularly apparent in the PW layer, where the
temperature difference between an upwelled parcel of water and that at the parcel's new neutral
buoyancy depth in the PW layer is greatest, and where iceberg melt rates (and therefore melt-driven
cooling) are generally smaller because of the low water temperatures. In contrast, when vertical





temperature gradients are shallower (e.g. with *AWcool*), cooling due to localised melting dominates
(blue lines in Fig. 7d,e and 7c). These effects tend to reduce vertical temperature variations of glacier-
adjacent waters compared both to simulations without icebergs and compared to conditions at the fjord
mouth.
Detailed near-glacier hydrographic observations against which to make comparisons are sparse, but
those that do exist provide some useful insight into the applicability of our model results to Greenland's
fjords. The pronounced surface and near-surface cooling (relative to conditions at the mouth) that we
simulate is a common feature in Greenland's fjords. For example, a transect of conductivity,
temperature, depth (CTD) casts along Sermilik Fjord revealed cooling of approximately 4°C in the
upper ~50 m (Straneo et al., 2011, 2012), which was also reproduced in a detailed modelling study of
Sermilik Fjord that included icebergs (Davison et al., 2020). Similar along-fjord near-surface cooling
has also been observed in other iceberg-congested fjords, such as Illulissat Isfjord (Beaird et al., 2017;
Gladish et al., 2015) and Upernavik Isfjord (Fenty et al., 2016), both in west Greenland. In Illulissat
Isfjord, the cold surface layer usually extends along-fjord to a shallow sill at the fjord mouth, where
icebergs frequently become grounded (Gladish et al., 2015).
Iceberg-induced changes to water properties below ~80 m are harder to identify in hydrographic
observations, most likely because they also contain the signature of glacial-plumes resulting from
subglacial discharge, or other external forcings. Our modelling suggests that, if vertical temperature
gradients are shallow, then icebergs can cause cooling over large depth ranges (e.g. Fig. 7c). As one
example, hydrographic observations in Kangerdlugssuaq Fjord showed relatively uniform near-glacier
temperatures with substantial cooling in both the upper 100 m and between 300 and 400 m depth,
relative to a transect acquired at the fjord mouth (Straneo et al., 2012), consistent with the modelling
results presented here. Iceberg-melt-induced warming of parts of the water column is harder to identify
in hydrographic observations because of the difficulty in distinguishing it from relatively warm
subglacial runoff-driven plume outflow.
To further compare our modelling results to observations, we examined CTD casts acquired as part of
the Oceans Melting Greenland (OMG) project (https://omg.jpl.nasa.gov/; data available at:
https://omg.jpl.nasa.gov/portal/browse/OMGEV-AXCTD/). As with the previous comparisons, and in
keeping with our simulation design, we selected pairs of CTD casts acquired less than a week apart,
one near or outside the fjord mouth and the other as close as possible to the tidewater glacier at the head
of the fjord. These profiles (Fig. 8) show many of the characteristics that we have simulated here.
Specifically, the profiles show that near-surface water temperatures are substantially colder adjacent to
tidewater glaciers compared to those observed outside each fjord, and the observed temperature
differences are comparable to those simulated here. In all but two (Illulissat Isfjord and Timmiarmiut
Fjord) of the surveyed fjords, the profiles also show warming at intermediate depths (~50-200 m)

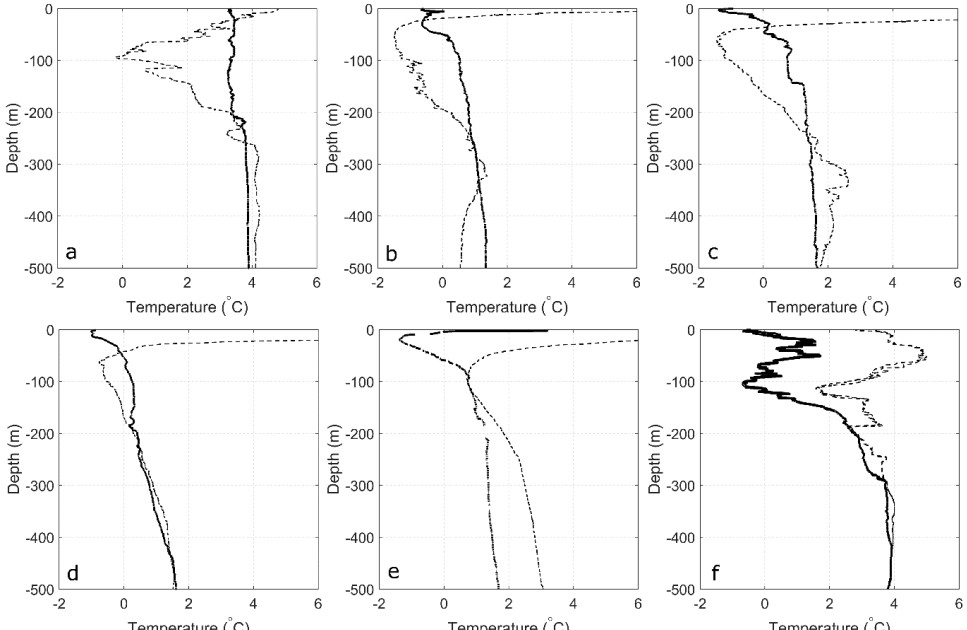

Figure 8. Fjord temperature profiles from the Oceans Melting Greenland project (https://omg.jpl.nasa.gov/). In panels, solid lines are profiles acquired within the fjord, close to tidewater glacier termini, and the dashed lines are acquired at or beyond the fjord mouth. Fjords (or nearest glacier) shown are (a) Sermilik Fjord, (b) Daugaard-Jensen, (c) Upernavik Isstrom, (d) Nunatakassaap Sermia Fjord, (e) Ilulissat Isfjord, and (f) Timmiarmiut Fjord. Note, in (f), both an up- and down-cast are shown for the outer part of the fjord. Data are available from: https://omg.jpl.nasa.gov/portal/browse/OMGEV-AXCTD/

relative to the waters outside the fjord. These observations do not allow us to quantify the relative
contributions to intermediate depth warming between plume outflow and iceberg melt-induced
upwelling. However, we note that the vertical pattern and magnitudes of intermediate depth warming
are similar to those simulated here. In addition, the intermediate depth warming occurs over a large
depth range, which is not easily explained by plume outflow and is consistent with our simulations.
Some of the profiles also show notable cooling at depth (e.g. Illulissat Isfjord), which we are only able
to reproduce in simulations including a shallow sill. Our simulations may underestimate cooling at
depth because power law size-frequency distributions underestimate the number of very large icebergs
(Sulak et al., 2017) and because the parameter values used in our melt calculation may underestimate
submarine melt rates (Jackson et al., 2020).

**4.2. Implications for glacier-ocean interaction**
If iceberg-induced changes to glacier-adjacent water properties significantly affect the magnitude
and/or the vertical pattern of glacier submarine melting, then icebergs may play an important role in

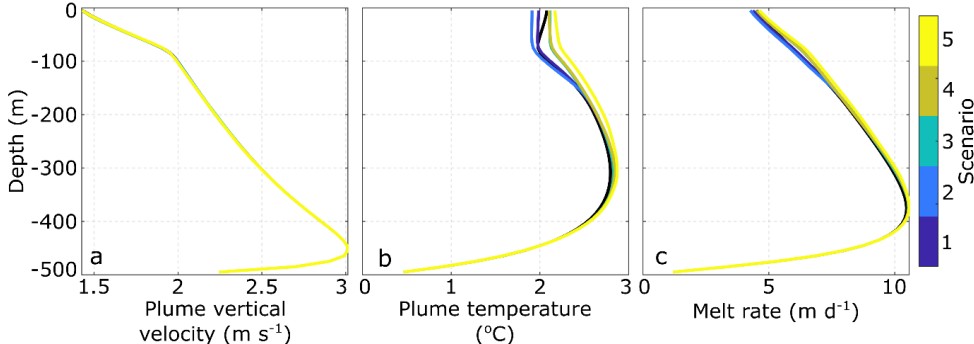

Figure 9. Plume dynamics for iceberg scenarios one to five. (a) Plume vertical velocity. (b) Plume temperature. (c) Glacier submarine melt rate in the plume. All simulations are based on *BCstandard* boundary conditions and 500 m s$^{-1}$ runoff.

modifying glacier response to ocean forcing. To assess the effect of icebergs on glacier submarine
melting, we first consider how iceberg-melt impacts subglacial runoff-driven plume dynamics and then
assess how the simulated temperature changes could affect melt rates across the parts of glacier fronts
that are not directly affected by runoff-driven plumes.
To examine the effect of icebergs on subglacial discharge plume-driven glacier submarine melting, we
evaluate plume properties for a single set of ocean boundary conditions (*BCstandard;* Fig. 1b-d) using
each of the five iceberg scenarios. We find that submarine iceberg melting has negligible influence on
plume vertical velocity and only modest influence on plume temperature, meaning plume-induced
glacier submarine melt rates appear relatively insensitive to the changes in temperature and salinity
induced by changes in iceberg geometry, concentration and size-frequency distribution (Fig. 9).
Although runoff-driven plume dynamics appear to be relatively insensitive to iceberg-induced
modification of glacier-adjacent water properties, submarine melting distal to glacial plumes
('background melting' (e.g. Slater et al., 2018)) may be more directly affected. Qualitatively, the
iceberg-melt-induced changes to glacier-adjacent water properties presented above suggest that iceberg
melt will affect background glacier melt rates in three key ways: (1) at and near the fjord surface, cooling
will reduce background melt rates; (2) in the PW layer, background melting will usually increase due
to upwelling of warmer AW, and; (3) in the AW layer, iceberg melt-induced changes in background
melt rates are expected to be modest, with slight increases in fjords with steep vertical temperature
gradients, and slight decreases in other fjords (assuming icebergs penetrate into the AW layer). These
effects will be more pronounced in fjords with higher concentrations of larger (and thus deeper keeled)
icebergs. In some fjords, then, where icebergs cause cooling near the surface and warming at depth, we
expect icebergs will increase glacier undercutting through impacting submarine melt rates, which may



382 in turn influence the rate and mechanism of calving (Benn et al., 2017; James et al., 2014; O'Leary and

383 Christoffersen, 2013).

384 To explore these effects quantitatively, we calculate the percentage change in background melt rate of

385 the glacier terminus due to iceberg-induced modification of glacier-adjacent water temperature (relative

386 to simulations without icebergs). Modelling studies indicate that background melt rates scale linearly

387 with ocean temperature (Sciascia et al., 2013; Slater et al., 2016; Xu et al., 2013); thus, changes in

388 temperature, $T$, should cause proportional changes in background melting (Jackson et al., 2014). We

389 choose to focus on relative changes in melt rate, rather than absolute changes, because of poor

390 constraints on important melt rate parameter values (Jackson et al., 2020). We calculate the relative

391 change in submarine melt rate, $SMR$, following Jackson et al. (2014), as:

392
$$\Delta SMR = \frac{(T_{ib} - T_f) - (T_{nib} - T_f)}{(T_{nib} - T_f)} 100$$

393 where the subscripts $ib$ and $nib$ indicate simulations with 'icebergs' and 'no icebergs', respectively, and

394 $T_f$ is the *in-situ* freezing point, given by:

395
$$T_f = \lambda_1 S + \lambda_2 + \lambda_3 z$$

396 where $\lambda_{1-3}$ are constants representing the freezing point slope (-0.0573 °C psu$^{-1}$), offset (0.0832°C) and

397 depth (0.000761°C m$^{-1}$), respectively. $S$ is the local salinity (horizontally averaged within 2 km of the

398 terminus) and $z$ is depth in the water column.

399 Using this approach, we find that the impact on water properties resulting from iceberg melt

400 substantially modifies background glacier submarine melt rates. Firstly, in the upper 50 m and using

401 *BCstandard*, iceberg melt causes a 34.9% reduction in melt rate on average. Even in iceberg scenario

402 one, iceberg melt causes a 29.5% reduction in melt rate over this depth range. Secondly, between 100

403 and 200 m depth, iceberg melt causes a 13.5% increase in melt rate on average when using *BCstandard*,

404 but this increases to 59.2% when using *PWcool* (for which warming of the PW layer due to upwelling

405 is most pronounced). Changes in iceberg melt rates in the AW layer are minimal, with the most

406 pronounced effect being a 5.4% increase in the 200-400 m depth range using iceberg scenario five and

407 *PWwarm*. When averaged through the entire water column, these effects largely compensate for each

408 other, resulting in a net 3.1% decrease in melt rates with *BCstandard*. Overall therefore, this analysis

409 suggests that iceberg melt can influence the vertical pattern of glacier terminus background melting by

410 decreasing melt rates at the surface and increasing them in the PW layer, with minimal changes in the

411 AW layer.

412 As well as affecting glacier-adjacent water temperatures, iceberg melt likely affects submarine melt

413 rates in other ways not examined here. For example, the cooling and freshening of the surface and near-

414 surface layers induced by iceberg melting may prevent or hinder plume surfacing (De Andrés et al.,



2020), and may expedite sea ice formation after the melt season, promoting the development of an ice
mélange. In addition, mechanical iceberg breakup, iceberg calving and iceberg rotation can cause
vigorous mixing of fjord waters which can temporarily increase glacier and iceberg submarine melt
rates (Enderlin et al., 2018), and increases the iceberg-ocean contact area available for melting. Iceberg-
melt-induced invigoration of fjord circulation can increase oceanic heat flux towards tidewater glaciers
(Davison et al., 2020), likely resulting in faster terminus submarine melting. Icebergs likely also exert
a mechanical influence on the circulation and plume dynamics at the ice-ocean interface (Amundson et
al., 2020), and may prevent plume surfacing (Xie et al., 2019).

**4.3. Implications for oceanic forcing of ice sheet-scale models**
Current state-of-the-art projections of dynamic mass loss from the Greenland Ice Sheet (Goelzer et al.,
2020) are forced by far-field ocean temperature profiles, provided by ocean modelling output that does
not include fjord-scale processes (except for the obstruction of shelf-water intrusion by shallow sills)
(Slater et al., 2019, 2020). The results presented here suggest that such an approach is broadly
appropriate for fjords with maximum iceberg keel depths of less than 200 m and iceberg concentrations
less than ~20% on average, where iceberg modification of glacier-adjacent water properties appears to
be limited other than in the upper several tens of metres (Fig.s 4 and 6). The majority of Greenland's
fjords likely fall into this category (Mankoff et al., 2019; Sulak et al., 2017). Even in such fjords,
however, this approach would not capture the surface and near-surface cooling caused by iceberg
melting. In order to capture this near surface cooling, one relatively simple modification to such an
approach could be to reduce surface water temperature to close to the *in-situ* melting point during winter
periods, and proportionally to the iceberg surface area at the fjord surface during summer periods.
However, in fjords hosting icebergs with keel depth greater than or equal to 200 m and with average
concentrations of more than ~20% (i.e. our iceberg scenario three or higher), iceberg modification of
glacier-adjacent water properties becomes increasingly important. In such fjords that also exhibit
relatively shallow sills, icebergs act to cool glacier-adjacent water throughout the water column, with
the amount of cooling proportional to the draught and concentration of the icebergs, as well as to the
temperature of the ambient water at the fjord mouth (Fig. 4). In such fjords that do not have shallow
sills, the effect is more complicated, with both iceberg-melt-induced warming and cooling, depending
on the vertical temperature gradient of the water column and iceberg concentration at depth. Overall,
these changes to the water column temperature can cause non-negligible (up to several tens of percent)
changes in terminus submarine melt rates across the large areas of the calving front that are not directly
affected by plume-inducing subglacial discharge. The vertical pattern of changes to terminus submarine
melt rates (reduced near the surface and increased at intermediate depths) induced by iceberg melting
is expected to exacerbate undercutting of glacier termini, with potentially important impacts on calving





rates (Benn et al., 2017; Ma and Bassis, 2019; O'Leary and Christoffersen, 2013; Todd and
Christoffersen, 2014). Although fjords hosting icebergs this large and numerous are relatively few in
number, it is these fjords (and the glaciers hosted by them) that contribute the most to dynamic mass
loss from the Greenland Ice Sheet (Enderlin et al., 2014; Khan et al., 2020).

**4.4. Transience vs steady-state**
All of the results presented here were extracted from the final ten days of simulations that were run to
a quasi-steady state (i.e. the variable of interest had stabilised). In our domains without sills, steady-
state of temperature and salinity was generally reached after just ten to thirty days. However, our
simulations with sills could take as many as one thousand days to reach such a steady state because
fjord-shelf exchange is reduced. For an equivalent steady-state to be reached in reality, open ocean
conditions, runoff and iceberg size and distribution would also have to remain quasi-stable for an
equivalent time period. In reality, this is unlikely to occur (particularly in fjords with shallow sills)
because runoff and coastal and open ocean conditions change on sub-seasonal to seasonal timescales
(Moon et al., 2017; Mortensen et al., 2014; Noël et al., 2016; Sutherland et al., 2014; Sutherland and
Pickart, 2008). In reality therefore, glacier-adjacent water properties in fjords with shallow sills are
likely a complex amalgamation of temporally-evolving source waters, modified by processes operating
within the fjord. In addition, some variations in coastal conditions can be transmitted towards glaciers
very rapidly. During winter, strong wind events on the east coast of Greenland drive fast shelf-forced
flows (or intermediary currents) in glacial fjords, delivering coastal waters to tidewater glaciers over
just a period of a few days, and potentially reducing the magnitude of iceberg-driven modification
(Jackson et al., 2014, 2018). Such currents are strongest in winter, when hydrographic observations are
sparse, so this remains speculative.

**5.   Conclusions**
We have used a general circulation model (MITgcm) to quantify the effect of submarine iceberg melting
on glacier-adjacent water properties in an idealised fjord domain. A large range of iceberg
concentrations, keel depths and size-frequency distributions were examined to represent the range of
iceberg conditions found in Greenland's marine terminating glacier fjords. We focused primarily on
iceberg-melt-induced changes to glacier-adjacent water temperatures throughout the water column,
because of their principal importance to glacier-submarine melting.
Our results suggest that icebergs can substantially modify glacier-adjacent water properties and that the
precise impact depends on iceberg size and on the temperature profile and stratification of water within
and beyond the fjord. In particular, we find that (1) temperature in the upper ~60 m of the water column
is reduced by several degrees Celsius over a wide range of iceberg scenarios; (2) fjords with more and



deeper icebergs are subject to greater iceberg-melt-induced modification, which can result in either
cooling or warming at different depths depending on the balance between melt-driven cooling and
upwelling-driven warming, which in turn depends on fjord temperature stratification, and; (3) when
icebergs extend to or below the fjord mouth sill depth, they can cause significant cooling throughout
the water column. Particularly with regard to point (2), our results highlight that oceanic forcing of large
fast-flowing glaciers, which contribute the most to ice sheet dynamic mass loss, in existing projections
of tidewater glacier dynamics is strongly affected by ignoring the impact of icebergs on fjord water
properties. The iceberg-induced changes to the vertical temperature profile of glacier-adjacent waters
identified here are likely to reduce submarine melt rates at and near the fjord surface while increasing
them in the PW layer, which may influence the rate and mechanism of calving by exacerbating glacier
terminus undercutting. Our results therefore identify a critical need to develop simple parameterisations
of iceberg-induced modification of fjord waters, and other fjord-scale processes, to better constrain
oceanic forcing of tidewater glaciers.


**Code availability**
MITgcm is freely available at http://mitgcm.org/public/source_code.html. The IcePlume module is
available    from    Tom    Cowton    on    request.    The    IceBerg    module    is    available    at
https://zenodo.org/record/3979647#.YWAayNrMKUk or from Benjamin Davison on request.

**Data availability**
Data required to reproduce the analysis and figures in this manuscript will be made available upon
publication.

**Author contributions**
BD and TC conceived the study. BD developed the model code with support from TC and AS. BD
designed and conducted the simulations and analysis, and led the manuscript write up. TC, FC, AS and
PN supported the interpretation of the model results and contributed to the preparation of the
manuscript.

**Competing interests**



The authors declare that they have no conflict of interest.

**Acknowledgements**
BD was funded by a PhD studentship provided by the Scottish Alliance for Geosciences, Environment
and Society (SAGES) and the University of St Andrews, UK. The simulations were conducted on the
Sheffield Advanced Research Computer (ShARC).

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
