# Peer review of "Modelling the effect of submarine iceberg melting on glacier-adjacent"

_The Cryosphere, 2021_

## Referee Comment (RC1)

Review for "Modelling the effect of submarine iceberg melting on glacier-adjacent water properties"
by Davison et al.
under consideration in The Cryosphere

Summary:
The authors describe a series of numerical modeling experiments that explore the influence of iceberg melting on ocean water properties at the glacier-ocean boundary. Modeling is performed using the MITgcm, which is widely used to simulate fjord circulation and ice-ocean interactions. The authors test a wide variety of iceberg conditions (including variations in keel depth, aerial coverage of icebergs, power-law distributions for iceberg sizes) as well as a number of different ocean temperature profiles, for idealized fjord geometries without and with a sill. The experiments are well thought-out, including covariance of the iceberg conditions, and likely span a variety of real-world conditions. The experiments indicate that iceberg melting causes ocean properties to become more uniform with depth. The homogenization of the water properties with depth should reduce glacier melting near the surface but increase glacier melting in the relatively cold near-surface Polar Water layer. These modifications to the glacier submarine melt profile may influence the stress balance at the glacier terminus, influencing terminus stability.

The manuscript is well written, interesting, novel, and easy to follow. A few questions regarding the applicability of the results given the limited fjord geometry and seasonality of the experiments, as well as some more minor comments are included below.

Major Points:
1.  For the iceberg melt parameterization, why were equations commonly used for glacier melting applied rather than the more traditional iceberg melt parameterizations from Bigg et al. (1997) and a number of more recent studies (Moon et al., 2018; Fitzmaurice et al., 2016; 2017)? Even if this choice is justified in Davison et al. (2020), it should be briefly explained here as well since it may strongly influence the melt rate estimates. It is not apparent why a plume-based model should be used when a face-normal (meaning horizontal for the vertical iceberg sides) relative velocity is used to estimate the melt rate.
2.  Although I imagine that the simulations may have taken a considerable amount of time to execute, I wonder why the authors did not perform a subset of the same experiments using different fjord geometries. The implications of the study have the potential to be much more broad if a few other simple geometries are incorporated into the analysis. For example, would the results be markedly different if the fjord was shallower (~200 m-deep), such that the subglacial plume was ejected into the Polar Water layer? Additionally, would the relaxation time change considerably depending on fjord width? If I had to prioritize, I'd be much more interested in the influence of fjord depth on the analysis than fjord length.

3. Similar to my comment above, the authors only consider summer ocean conditions despite incorporating simulations without runoff. Winter hydrographic data are limited, as stated by the authors in the discussion, but some data are available in Sermilik Fjord as an example. The runoff vs no runoff portion of the manuscript is not really discussed beyond the results but this may be very important: it looks as though iceberg melt in the upper-most 50 m is entirely suppressed when runoff is present because the temperature near the surface is at the freezing point. This result suggests that melting at depth in the winter may buffer small icebergs from melting, promoting the growth of sea ice and mélange. Of course your model cannot yield insights into the influence of near-surface melt suppression on mélange properties, but it is certainly worth discussing.

Minor Comments:
- Table 1 comes before you explain the different scenarios, resulting in some confusion when the different iceberg configurations are described at the bottom of page 6. Consider moving this table or making it clear earlier-on that you modify a number of iceberg parameters separately and also in combination (if I am correctly interpreting the present description). Alternatively, you can omit the fact that you modified the parameters separately since you never discuss those independent modifications.
- Figure 2: How did you distribute the iceberg sizes across the fjord domain? They clearly are not uniformly-distributed across the fjord but there is no description of the distribution in the text.
- Make sure you are consistent with terminology. In the results, you describe simulations with and without subglacial discharge but the term runoff is used in the Figure 3 caption.
- line 333: Change "Iceberg-melt-induced" to "Iceberg melt-induced"
- Figure 8: I recommend averaging or down-sampling the observed profiles to the same depth resolution of the model simulations. It may also help to show the most similar profiles from the simulations in each plot.

---

## Author Comment (AC1)

Review for "Modelling the effect of submarine iceberg melting on glacier-adjacent water properties"
by Davison et al.
under consideration in The Cryosphere

**Summary:**
The authors describe a series of numerical modeling experiments that explore the influence of iceberg melting on ocean water properties at the glacier-ocean boundary. Modeling is performed using the MITgcm, which is widely used to simulate fjord circulation and ice ocean interactions. The authors test a wide variety of iceberg conditions (including variations in keel depth, aerial coverage of icebergs, power-law distributions for iceberg sizes) as well as a number of different ocean temperature profiles, for idealized fjord geometries without and with a sill. The experiments are well thought-out, including covariance of the iceberg conditions, and likely span a variety of real-world conditions. The experiments indicate that iceberg melting causes ocean properties to become more uniform with depth. The homogenization of the water properties with depth should reduce glacier melting near the surface but increase glacier melting in the relatively cold near-surface Polar Water layer. These modifications to the glacier submarine melt profile may influence the stress balance at the glacier terminus, influencing terminus stability.

The manuscript is well written, interesting, novel, and easy to follow. A few questions regarding the applicability of the results given the limited fjord geometry and seasonality of the experiments, as well as some more minor comments are included below.

We thank the reviewer for providing a thorough and supportive review of our manuscript. We have addressed all of their questions and suggestions. In most cases, we agreed with and have implemented their suggestions. The reviewer suggested that additional simulations examining fjord depth and seasonality should be carried out. We have chosen not to undertake these simulations because we do not believe that the significant additional computational expense required is justified by the comparatively minor additional insight that these would provide, and argue based on our idealised simulations and other published high-fidelity simulations that they would not affect our conclusions. We have provided further reasoning in our response below.

**Major Points:**
1.  For the iceberg melt parameterization, why were equations commonly used for glacier melting applied rather than the more traditional iceberg melt parameterizations from Bigg et al. (1997) and a number of more recent studies (Moon et al., 2018; Fitzmaurice et al., 2016; 2017)? Even if this choice is justified in Davison et al. (2020), it should be briefly explained here as well since it may strongly influence the melt rate estimates. It is not apparent why a plume-based model should be used when a face-normal (meaning horizontal for the vertical iceberg sides) relative velocity is used to estimate the melt rate.

    The three-equation formulation (3EF) used here describes the thermodynamical equilibrium at an ice-ocean interface, and so its use need not be limited to calculating glacier submarine melting. The advantage of the 3EF over other iceberg parameterisations is that it is better suited for resolving vertical variations in melt rates because most other iceberg melt parameterisations were designed to calculate the so-called 'bulk' melt rate of an iceberg. We have modified our wording in the manuscript to emphasise this: "We chose to use this melt rate parameterisation, rather than existing iceberg melt

parameterisations (e.g. Bigg et al., 1997), because it enables us to resolve the vertical pattern of submarine melting…" (lines 130-132 in the revised manuscript).

Although the 3EF has in the past been coupled with a plume model to calculate plume-driven submarine melt rates, it is not in itself a plume-based model. Indeed, it has traditionally been used to calculate melt rates at the base of ice shelves, and was only subsequently applied to vertical glacier calving fronts. It is therefore appropriate to use the 3EF to calculate the melt rates of both the horizontal and vertical faces of the iceberg, as long as the relevant current velocity is used (which is actually the magnitude of the face-parallel and vertical velocity along the iceberg sides, and magnitude of both the horizontal velocities at the iceberg base, whilst accounting for iceberg drift). We did use a plume model to inform the choice of background velocity along the iceberg sides. This is justifiable because in that case, we are trying to parameterise the effect on melt of plumes rising along the iceberg face.

2.  Although I imagine that the simulations may have taken a considerable amount of time to execute, I wonder why the authors did not perform a subset of the same experiments using different fjord geometries. The implications of the study have the potential to be much more broad if a few other simple geometries are incorporated into the analysis. For example, would the results be markedly different if the fjord was shallower (~200 m deep), such that the subglacial plume was ejected into the Polar Water layer?

    Additionally, would the relaxation time change considerably depending on fjord width? If I had to prioritize, I'd be much more interested in the influence of fjord depth on the analysis than fjord length.

    The reviewer is correct that the simulations are computationally expensive, and so we tried to reduce the number of variables to test.

    In our simulations, the plume reaches the fjord surface, so reducing the depth of the fjord would not cause the plume to terminate in the Polar Water layer (i.e. lower in the water column). If we either increased the depth of the fjord or changed the boundary conditions to create a stronger density gradient at the Polar Water-Atlantic Water interface, it would be possible to make the plume terminate at that interface, and this would cause some additional warming of the Polar Water layer. This would accentuate the pattern found in the simulations already presented: cooling of the surface layer, warming of the Polar Water layer, and minimal changes to the Atlantic Water layer.

    Similarly, if a shallower fjord geometry was used, such that grounding line was at the depth of the Polar Water layer, then the runoff would be ejected directly into the Polar Water layer. Carroll et al (2016) demonstrated that in fjords with this geometry, plume-outflow is relatively cool and plume-driven glacier melt rates are relatively uniform with depth, except near the surface if the surface layer is seasonally warmed. With this geometry, there would be no warming of the Polar Water layer (because there is no warmer water below to mix vertically), and the relatively warm surface layer would be further cooled by iceberg melting. Hence both plume outflow and iceberg melting would cause the properties of the upper layer to tend towards those of the Polar Water layer, leading to more homogenous glacier submarine melt rates (in this case leading to reduced overcutting).

In answer to the second part of the reviewer's comment, the simulation wall clock time would decrease approximately linearly with fjord width. However, this would necessitate re-evaluating the iceberg geometries considerably, because there would be less space to host the large icebergs. For example, it would make designing the domain for a deep fjord, where large icebergs might be expected, very challenging. We chose to use a 5 km wide and 50 km long fjord because this is relevant to many fjords in Greenland and because it permits us to use a wide range of iceberg geometries without encountering the issue of choking the fjord with ice.

Carroll, D., Sutherland, D. A., Hudson, B., Moon, T., Catania, G. A., Shroyer, E. L., *et al*. 2016. The impact of glacier geometry on meltwater plume structure and submarine melt in Greenland fjords. *Geophysical Research Letters*, 43(18), 9739-9748. doi: 10.1002/2016GL070170.

3. Similar to my comment above, the authors only consider summer ocean conditions despite incorporating simulations without runoff. Winter hydrographic data are limited, as stated by the authors in the discussion, but some data are available in Sermilik Fjord as an example. The runoff vs no runoff portion of the manuscript is not really discussed beyond the results but this may be very important: it looks as though iceberg melt in the upper-most 50 m is entirely suppressed when runoff is present because the temperature near the surface is at the freezing point. This result suggests that melting at depth in the winter may buffer small icebergs from melting, promoting the growth of sea ice and mélange. Of course your model cannot yield insights into the influence of near-surface melt suppression on mélange properties, but it is certainly worth discussing.

As the reviewer points out, winter observations are limited. To the best of our knowledge, the only winter observations are two casts acquired within the mélange close to the Helheim glacier calving front, and two casts acquired ~50 km from the calving front, all in March 2010. However, there are no coincident observations at or beyond the fjord mouth that could be used at the open boundary, and so these observations are not suitable for the analysis presented in the manuscript. Nevertheless, we chose to include the results from simulations without runoff principally because they provide information regarding the effect of icebergs melting on near-glacier water properties, and help the reader to interpret the results from simulations including both runoff and icebergs.

The iceberg-induced cooling that we simulate does potentially indicate that iceberg melting pre-conditions the fjord for mélange formation during the winter, because the cold and relatively fresh surface will encourage sea ice formation and because icebergs will be more likely to persist through the winter, though our model cannot show this, as the reviewer points out.

The statement that 'iceberg melt in the upper-most 50 m is entirely suppressed when runoff is present because the temperature near the surface is at the freezing point' is not correct' – runoff increases iceberg melt rates, rather than supresses them. The relatively warm outflow from the plume does offset some (but not all) of the iceberg melt-induced cooling in the upper 50 m, which is perhaps the meaning that the reviewer intended.

Minor Comments:

- Table 1 comes before you explain the different scenarios, resulting in some confusion when the different iceberg configurations are described at the bottom of page 6. Done
- Consider moving this table or making it clear earlier-on that you modify a number of iceberg parameters separately and also in combination (if I am correctly interpreting the present description). Alternatively, you can omit the fact that you modified the parameters separately since you never discuss those independent modifications. We have moved the table to a more appropriate location.
- Figure 2: How did you distribute the iceberg sizes across the fjord domain? They clearly are not uniformly-distributed across the fjord but there is no description of the distribution in the text. We have clarified this on lines 162-164 of the revised manuscript: "In these setups, iceberg concentration decreases linearly in the along-fjord direction between specified maximum and minimum values (Table 1) and icebergs are distributed randomly in the across-fjord direction (Fig. 2)."
- Make sure you are consistent with terminology. In the results, you describe simulations with and without subglacial discharge but the term runoff is used in the Figure 3 caption. We have revised the caption to Figure 3 using "subglacial discharge" instead of "runoff".
- line 333: Change "Iceberg-melt-induced" to "Iceberg melt-induced" Done here and throughout the revised manuscript.
- Figure 8: I recommend averaging or down-sampling the observed profiles to the same depth resolution of the model simulations. It may also help to show the most similar profiles from the simulations in each plot. The revised version of this figure shows down-sampled versions of the observed profiles as the reviewer suggested. We have chosen not to plot simulated profiles on the same figure because, although our simulations show vertical patterns of water column temperature, our model boundary conditions are sufficiently different from some of the observed mouth profiles that the absolute temperature in the simulations differs from that observed in the fjord.

---

## Author Comment (AC2)

Review for „Modelling the effect of submarine iceberg melting on glacier-adjacent water properties" by Benjamin Davison and co-authors

The paper by Benjamin Davison and co-authors combines a general circulation ocean model (MITgcm) with a submarine iceberg melting module, in order to investigate the impact of iceberg melting and cooling, and their vertical distribution, on fjord water properties close to Arctic glaciers. The work is to my understanding an extension of the 2020 study by Davison et al. that was focused on a single Greenlandic tidewater glacier, by studying different likely iceberg "scenarios" and different simplified geometric fjord configurations in order to cover the wide range of Greenland fjord configurations. The work has potential implications also for efforts to project Greenland Ice Sheet behaviour with models, which usually can make use only of far-field properties beyond the fjord's mouth to force these ice sheet/ice shelf components, so the scientific relevance of the study is high; and the paper should, in my opinion, be published soon.

I think that the model setups are defined very elegantly in order to answer how iceberg melting affects glacier-adjacent water properties. While almost all simulations show a cooling in the upper 60m or so, below that level either warming or cooling can occur depending on the "icescape" and configuration.

Specifically, the paper implies that projections for the large fast-flowing Greenland glaciers that contribute most negatively to the mass balance are potentially affected by the lack of iceberg effects on fjord water properties (hosting numerous and large icebergs), and this is very clearly shown with simple model configurations. These "details" can potentially matter a lot for the "large-scale" mass balance of Greenland. Notably, the authors even provide a first idea for simple parameterizations (l.434-436) in their paper and I hope that these ideas will be picked up in the community promptly.

The last paragraph of section 4.1 attempts a comparison to observations with some success. Here, it would have been great to close the circle by saying more clearly (or even plotting in the same panels) which simple six model configurations can mirror panels a-f in Figure 8. Or in other words, which assumptions are needed to model profiles similar to the observed profiles (e.g. presence of a sill) with the simple fjord geometry used in the study.

The paper is written and organized excellently (with no obvious typos, which is rare), and the arguments are easy to follow. All results are clearly described and discussed and the conclusions are based entirely on the model results. The quality of the figures is also okay. Below, you can find a short list of line-by-line comments that the authors could still work on. I suggest to accept the paper with (very) minor revisions.

Thomas Rackow

We thank the reviewer for providing a thorough and supportive review of our manuscript. We agree with all of their comments and have implemented all of their suggestions in the revised version of the manuscript.

##################

Line-by-line comments:

l.89/90 Here or somewhere else, I think the high-impact study by Schaffer et al. (2020) should be mentioned who conclude that near-glacier sill-controlled ocean heat transport can play a crucial role for glacier stability.

Reference:

Schaffer, J., Kanzow, T., von Appen, WJ. et al. Bathymetry constrains ocean heat supply to Greenland's largest glacier tongue. Nat. Geosci. 13, 227–231 (2020). https://doi.org/10.1038/s41561-019-0529-x

Done.

l.94-109 What is the surface boundary condition at the atmosphere-ocean interface? I missed that somehow and it would be good to know whether this could influence the surface representation of the profiles (e.g. holding them close to some value).

The reviewer is correct to point out that the atmosphere-ocean boundary could influence the surface ocean conditions. In our simulations, we chose not to include any atmosphere-ocean interaction, so as to isolate the effect of icebergs on the fjord conditions. We mention this briefly on line 109 of the original manuscript (also line 109 in the revised manuscript): "…do not simulate the effects of sea ice, atmospheric forcing or tides". Although not realistic, this is in keeping with the approach of many other Greenland-focused fjord modelling studies (e.g. Cowton et al., 2015, 2016; Carroll et al., 2017) and allows us to more easily isolate the effect of icebergs melting on ocean conditions. We appreciate that some authors have chosen to include atmosphere-ocean interactions (e.g. Fraser et al., 2018), and that there may be interactions between the atmosphere-ocean interactions and that effect of iceberg melt on water properties that our simulations will not capture.

References:

Cowton, T., Slater, D., Sole, A., Goldberg, D., and Nienow, P. 2015. Modeling the impact of glacial runoff on fjord circulation and submarine melt rate using a new subgrid-scale parameterization for glacial plumes. *Journal of Geophysical Research: Oceans*, 120, 796-812. doi:10.1002/2014JC010324.

Cowton, T., Sole, A., Nienow, P., Slater, D., Wilton, D., and Hanna, E. Controls on the transport of oceanic heat to Kangerdlugssuaq Glacier, East Greenland. 2016. *Journal of Glaciology*. Doi: 10.1017/jog.2016.117

Carroll, D., Sutherland, D. A., Shroyer, E. L., Nash. J. D., Catania, G. A., and Stearns, L. A. 2017. Subglacial discharge-driven renewal of tidewater glacier fjords. *Journal of Geophysical Research: Oceans*, 122, 6611-6629. https://doi.org/10.1002/2017JC012962

Fraser, N. J., Inall, M. E., Magaldi, M. G., Haine, T. W. N., Jones, S. C. 2018. Wintertime fjord-shelf interaction and ice sheet melting in southeast Greenland. *Journal of Geophysical Research: Oceans*, 123, 9156-9166. https://doi.org/10.1029/2018JC014435

l.129/130 I think it would be good to also cite the much earlier Hellmer & Olbers (1989) study for the three-equation formulation, which is often forgotten:

Hellmer, H., and D. Olbers (1989), A two-dimensional model for the thermohaline circulation under an ice shelf, Antarct. Sci., 1, 325–336, doi:10.1017/S0954102089000490.

Agreed and done.

l.396-398 The constants are also from Jackson et al. 2020?

These constants for calculating the freezing point were taken from Cowton et al. (2015), which in turn were based on those originally presented in Holland and Jenkins (1999). The exact value for the constants used do vary slightly between publications, though many papers do not provide the values used. We now cite Cowton et al. (2015) on line 397 (in both the original and revised manuscript).

Holland, D. M. and Jenkins, A.: Modeling Thermodynamic Ice–Ocean Interactions at the Base of an Ice Shelf, J. Phys. Oceanogr., 29(8), 1787–1800, doi:10.1175/1520-0485(1999)029<1787:MTIOIA>2.0.CO;2, 1999.

Section 4.2: You tried to explore the relative change in submarine melt rate quantitatively, which is great. To my understanding, this is a simple diagnostic and the model does not see the different melt rates. I was wondering whether any feedbacks are to be expected, or whether your conclusions might be different in a model setup that would account for the iceberg-induced melt changes?

In our simulations, we used 'IcePlume' to simulate melting of the calving front. IcePlume does simulate submarine melting in areas distal to the runoff-driven plume, and these are affected by the inclusion of icebergs in the domain due to the iceberg-induced changes to the water column temperature. However, parameterisations of submarine melting in these regions is extremely uncertain (e.g. Jackson et al., 2020), which is why we chose to use the relative method of Jackon et al. (2014). The relative method does assume that all changes in temperature affect submarine melt rates (i.e. if an increase in temperature increases heat supply to the glacier face, it assumes that all of that heat supply is used in submarine melting). In reality, this is likely an upper-bound on the effect of temperature changes on glacier submarine melt rates – we have modified the wording in the revised manuscript to reflect this: "It is worth noting that changes in melt rate calculated using this method assume that all changes in heat supply are accommodated by changes in submarine melt rates, and so this method provides an indication of the maximum relative changes in submarine melt rates expected due to changes in ambient ocean temperature" (line 398-401 in the revised manuscript)

We don't expect there to be strong feedbacks between glacier melting and fjord circulation associated with iceberg-induced changes to the glacier submarine melt profile. This is because the additional volume flux of meltwater from those portions of the glacier experiencing accelerated submarine melting is small in comparison to that provided from runoff and iceberg melting. For example, glacier submarine melt rates in these regions are thought to be around 0.5 metres per day. There's uncertainty in these values, so let's suppose a maximum value of 3 metres per day. We find, at most, a 60% increase in melt rates in the 100-200 m depth range, which equates to a volume flux of ~10 $m^3$ $s^{-1}$ when distributed over a 5 km-wide ice wall. The water motion driven by this freshwater flux might act to slightly increase melt rates higher in the water column, but we suspect that more powerful currents driven by plumes, iceberg melting and tides, for example, would make a positive feedback unlikely. Put more succinctly: although the melt rates of the glacier face are similar to that of the icebergs, the submerged area of the calving front is a small fraction of the submerged iceberg area, so the impact on the fjord circulation due to small changes in glacier submarine melt rates is likely to be proportionally small.